# Adenosine 2B Receptor Signaling Impairs Vaccine-Mediated Protection Against Pneumococcal Infection in Young Hosts by Blunting Neutrophil Killing of Antibody-Opsonized Bacteria

**DOI:** 10.3390/vaccines13040414

**Published:** 2025-04-15

**Authors:** Shaunna R. Simmons, Alexsandra P. Lenhard, Michael C. Battaglia, Elsa N. Bou Ghanem

**Affiliations:** Department of Microbiology and Immunology, Jacobs School of Medicine and Biomedical Sciences, University at Buffalo, Buffalo, NY 14203, USA; ss627@buffalo.edu (S.R.S.); apabamon@buffalo.edu (A.P.L.); mcbattag@buffalo.edu (M.C.B.)

**Keywords:** pneumococcal conjugate vaccine, PMNs, extracellular adenosine, aging

## Abstract

**Background/Objective:** Neutrophils are essential for vaccine-mediated protection against pneumococcal infection and impairment in their antibacterial function contributes to reduced vaccine efficacy during aging. However, the signaling pathways that control the neutrophil responses in vaccinated hosts are not fully understood. The extracellular adenosine pathway is a known regulator of neutrophils in naïve hosts. The aim of this study was to test the role of this pathway in the function of neutrophils and their protection against infection upon vaccination as a function of the host’s age. **Methods:** To test the role of adenosine in the antimicrobial activity of neutrophils against antibody-opsonized pneumococci, we used bone marrow-derived neutrophils isolated from wild-type or specific-adenosine-receptors knock-out mice. To measure the effect of adenosine receptor signaling in vivo, we treated vaccinated mice with agonists or antagonists that were specific to the different adenosine receptors prior to pulmonary challenge with pneumococci and assessed the bacterial burden and clinical score post-infection. **Results:** We found that signaling via the adenosine 2B (A2BR) receptor but not the A2A or A1 receptors diminished the intracellular pneumococcal killing following antibody-mediated uptake in young hosts. In vivo, the agonism of the A2BR receptor significantly worsened the pneumococcal infection outcomes in young, vaccinated mice. In contrast, A2BR signaling had no effect on the intracellular bacterial killing by neutrophils from aged mice. Further, in vivo A2BR inhibition had no effect on the pneumococcal disease progression in aged, vaccinated mice. **Conclusions:** A2BR signaling reduced pneumococcal vaccine-mediated protection by impairing the antimicrobial activity of neutrophils against antibody-opsonized bacteria in young hosts. However, inhibiting this pathway was not sufficient to boost responses in aged hosts.

## 1. Introduction

*Streptococcus pneumoniae* (pneumococcus) are Gram-positive, encapsulated bacteria that are the leading cause of bacterial pneumonia globally, and the leading cause of community-acquired bacterial pneumonia in those over the age of 65 [1,2,3,4]. Older adults are at an increased risk of contracting pneumococcal disease, with an increased number of cases and an increased mortality rate occurring in those over the age of 65 [5]. The increased susceptibility of older adults to pneumococcal pneumonia is due to declines in both the adaptive and innate immune responses that accompany aging, known as immunosenescence [6,7]. The increased risk of pneumococcal pneumonia persists in this population despite the availability of vaccines [5]. Licensed pneumococcal vaccines include the pneumococcal conjugate vaccine (PCV), which is currently recommended for older adults [8]. These vaccines target bacterial polysaccharides but have reduced efficacy in protection against pneumococcal pneumonia in older adults [9,10,11]. Reduced vaccine efficacy with age can be partly attributed to immunosenescence [11]. Vaccination helps mediate pneumococcal bacterial clearance in a variety of ways; antibody binding on the surface of bacteria prevents the bacteria from binding to lung epithelia and can induce complement-mediated killing [12]. Additionally, opsonization, or coating of the bacterial surface with opsonins such as complement proteins or antibodies, can induce phagocytosis. In a vaccinated individual, the opsonization of the pneumococcus by antibodies promotes uptake and clearance by phagocytes, including neutrophils, also known as polymorphonuclear leukocytes (PMNs) [11,12]. However, the number and function of anti-pneumococcal antibodies and neutrophil functionality decline with host aging [11,13,14].

PMNs are important mediators of pneumococcal disease outcome and are one of the first cells recruited to the lung following the occurrence pneumococcal pneumonia [7,15]. PMNs kill bacteria in a variety of ways, including phagocytosis and subsequent intracellular bacterial killing, degranulation and the release of preformed antimicrobial products, the production of reactive oxygen species (ROS), and through the release of neutrophil extracellular traps (NETs) [16]. PMNs are essential for host defense against pneumonia; neutropenic patients are at higher risk of contracting pneumonia and, in mouse models of pneumococcal infection, the depletion of PMNs prior to pulmonary challenge resulted in a significantly higher bacterial burden and mortality [1,13,15,17,18]. We also showed that PMNs are important for the protection of vaccinated hosts, where the depletion of PMNs in PCV-immunized mice resulted in a significant reduction in host survival upon pulmonary challenge with *S. pneumoniae* [13]. We found that the antimicrobial activity of PMNs against antibody-opsonized bacteria declined in vaccinated aged hosts and that this decline persisted even when antibodies were isolated from young controls, suggesting an intrinsic defect in the function of the PMNs [13]. Following antibody-mediated uptake, there was a significant defect in the intracellular pneumococcal killing by PMNs isolated from aged mice. The adoptive transfer of PMNs from young, naïve mice into aged, vaccinated mice significantly enhanced the host protection following pneumococcal infection. These data show that improved PMN function in aged, vaccinated hosts can help to improve pneumococcal disease outcome [13].

A known regulator of immune cell function is the extracellular adenosine (EAD) pathway. During infection and inflammation, ATP leaks from damaged cells and is converted to extracellular adenosine by dephosphorylation by the extracellular enzymes CD39 and CD73 [19,20]. EAD can then act on the four G-protein-coupled adenosine receptors, A1, A2A, A2B, and A3, that can have opposing downstream effects on immune cell responses [19,20]. Adenosine receptors are expressed on PMNs and this pathway has been shown to regulate the responses of PMNs to bacterial infection, or to stimulation by bacterial products [20]. For example, A1 receptor (A1R) signaling was found to be necessary for efficient PMN influx to the lungs following pneumococcal infection [21] and was also required for the efficient intracellular killing of *S. pneumoniae* by PMNs [22]. Activation of the A2A receptor (A2AR) was shown to decrease the MMP-9 secretion by fMLP-activated PMNs [23,24]. Further, 2B receptor (A2BR) knock-out mice had improved survival upon pulmonary infection with *Klebsiella pneumoniae* and A2BR^−/−^ PMNs had higher production of NETs and increased bactericidal activity compared to wild-type PMNs [25]. Similarly, A2BR^−/−^ mice were more resistant to pneumococcal infection and A2BR^−/−^ PMNs had increased mitochondrial ROS production that resulted in increased bactericidal activity against *S. pneumoniae* [26,27].

Although EAD has been shown to regulate the PMN response in naïve hosts, our understanding of how EAD controls antimicrobial response of PMNs in immune hosts remains limited. The way that PMNs interact with bacteria in naïve versus vaccinated hosts can significantly differ due to the differences in opsonins. The pneumococcal capsule helps bacteria evade phagocytosis [4]; however, pneumococci can be opsonized by complement or antibody deposition on the bacterial surface, which helps promote phagocytosis and clearance of the bacteria via the activation of complement receptors (CRs) and Fc receptors on the neutrophil surface [4,28,29]. It is important to understand the differences between complement-mediated and antibody-mediated responses by PMNs because the activation of theses receptors results in distinct signaling pathway activation in PMNs [28,30,31]. It was shown that the phagocytosis of complement-coated or IgG-coated beads by PMNs induced differing levels of ROS, differences in phagocytosis, and receptor-specific changes in gene expression [32]. To determine the pathways controlling the PMN responses in a vaccinated host, we asked if EAD signaling regulates the killing of *S. pneumoniae* following antibody-mediated uptake by PMNs and whether this pathway can be targeted to reverse the decline in vaccine protection in aged hosts.

## 2. Materials and Methods

### 2.1. Mice

Young (2–3 months) and old (18–22 months) C57BL/6 (B6) male mice were either purchased from Jackson Laboratory (C57BL/6J) or obtained from the National Institute on Aging colony housed at Charles River facilities (C57BL/6JN). A2BR^−/−^ on a B6 background (B6.129P2-Adora2btm1Till/J) and A2AR^−/−^ mice on a Balb/c background (C;129S-Adora2atm1Jfc/J) were purchased from Jackson Laboratory and bred at our facility. Age-matched wild-type (WT) C57BL/6J and Balb/cJ mice were used as controls. All mice were housed in specific pathogen-free housing for at least two weeks prior to use in experiments. A2BR^−/−^ mice bred at our facility were aged for 18 months.

### 2.2. Ethics Statement

All animal studies were performed in accordance with the recommendations in the Guide for the Care and Use of Laboratory Animals and in accordance with the University at Buffalo Institutional Animal Care and Use Committee guidelines.

### 2.3. Bacteria

*Streptococcus pneumoniae* TIGR4 strains (serotype 4), wild-type (WT) and pneumolysin deletion mutant (*Δply*), were a gift from Andrew Camilli [33]. Bacteria were grown to mid-exponential phase at 37 °C and 5% CO_2_ in Todd Hewitt broth supplemented with 0.5% yeast extract and oxyrase, as previously described [34].

### 2.4. PMN Isolation

Bone marrow (BM) was isolated from the leg bones of naïve, uninfected mice by flushing the bones with Roswell Park Memorial Institute 1640 media (RPMI) supplemented with 10% fetal bovine serum (FBS) and 2mM ethylenediaminetetraacetic acid (EDTA) (Fisher S31200). This BM suspension was then strained using 100 μm cell strainers and red blood cells were lysed. Cells were washed and resuspended in phosphate-buffered saline (PBS). PMNs were isolated via density centrifugation using Histopaque 1119 and 1077 (Sigma), as described before [35]. PMNs were resuspended at the indicated concentrations for each experiment in Hank’s Balanced Salt Solution (HBSS)/0.1% gelatin with no Ca^2+^ or Mg^2+^ (Gibco, Grand Island, NY, USA 14-175-103) and placed on ice until use. PMN purity was confirmed by flow cytometry. It was determined that 85–90% of the cells were Ly6G and CD11b positive.

### 2.5. Generation of Immune Sera

Young WT mice were vaccinated intramuscularly into the caudal thigh muscle with 50 μL of the pneumococcal conjugate vaccine (PCV) Prevnar-13^®^ (Wyeth pharmaceuticals, Madison, NJ, USA). Four weeks following vaccination, mice were euthanized, and blood was collected via portal vein snip and centrifuged at 9000 rpm to separate the sera. Sera were then stored at −80 °C until use. To heat-inactivate (HI) the complement proteins in the immune sera prior to use, sera were incubated at 56 ⁰C for 40 min as previously described [36]. We previously demonstrated that heat inactivation prevented complement deposition but had no effect on anti-capsular IgG binding to bacterial surfaces [13].

### 2.6. Mouse Infection and Treatment

Mice were vaccinated with Prevnar-13^®^ (PCV-13) four weeks prior to treatment and infection. Mice were infected with 50 μL of *S. pneumoniae* TIGR4 at concentrations indicated via assisted aspiration, which we refer to here as intratracheally (i.t.). Briefly, mice were anesthetized using isoflurane and pneumococcus was delivered to the lungs by pipetting bacteria into the trachea with the tongue gently pulled to the side. For treatment with specific adenosine receptor agonists and antagonists, mice were injected intraperitoneally (i.p) 18 h prior to, at the time of, and 18 h post-infection. Young mice were treated with specific A2B receptor agonist Bay 60-6583 (Tocris Bioscience, Bristol, UK) dissolved in sterile dimethyl sulfoxide (DMSO) (Fisher, Hampton, NH, USA, 31765ML), which was filter sterilized by being passed through a 0.22 μm filter prior to use and was given to mice via intraperitoneal injection at a volume of 100 μL and a concentration of 2 mg/kg. Old mice were treated with specific A2B receptor antagonist MRS 1754 (Tocris Bioscience) dissolved in DMSO, which was filter sterilized by being passed through a 0.22 μm filter prior to use and was given to mice via intraperitoneal injection at a volume of 100 μL and a concentration of 2.5 mg/kg. Where indicated, old mice were also treated with specific A2A receptor antagonist 3,7-Dimethyl-1-propargylxanthine (Sigma, St. Louis, MI, USA) dissolved in DMSO, which was filter sterilized by being passed through a 0.22 μm filter prior to use and was given to mice via intraperitoneal injection (100 μL) at a concentration of 5 mg/kg simultaneously with treatment of A2BR antagonist MRS 1754. Control mice were treated with diluted DMSO as vehicle control. Twenty-four hours post-infection, clinical signs of disease (clinical score) were determined for each mouse. Mice were then euthanized, and blood was collected to determine levels of bacteremia. Following blood collection, mice were perfused with sterile PBS through the heart and then lungs and brain were harvested. Lung and brain tissues were homogenized in sterile PBS, diluted in PBS, and plated on blood agar to determine colony-forming units (CFU) in each organ.

### 2.7. Gentamicin Protection Assay

PMNs were isolated from the bone marrow of unvaccinated mice as indicated, as we previously found that vaccination did not alter intrinsic PMN antimicrobial activity against *S. pneumoniae* [13]. PMNs were then infected at a MOI of 25 with *Δply S. pneumoniae* TIGR4, and were pre-opsonized with 3% naïve or heat-inactivated immune sera for 10 min at 37 °C. Next, Gentamicin (100 μg/mL) (Fisher, 15710064) was added for 30 min to kill any extracellular bacteria that were not taken in by the PMNs. PMNs were subsequently washed and resuspended in HBSS/0.1% gelatin. Half of the reaction was diluted and immediately plated on blood agar plates to determine the amounts of intracellular bacteria. The other half of the PMNs were incubated for 15 more minutes at 37 °C and then plated on blood agar to determine the amount of remaining viable intracellular bacteria. We then calculated the percentage of engulfed bacteria that was killed. We previously found no significant differences between PMNs’ overall killing of wild-type and *Δply S. pneumoniae* TIGR4, but the presence of the pore-forming toxin pneumolysin (PLY) prevented the detection of intracellular bacteria using gentamicin protection, as the pores allowed influx of the antibiotic into the PMNs [34]. Where indicated, PMNs were pretreated for 30 min at 37 °C with the specific A1 agonist 2-Chloro-N6-cyclopentyladenosine (Tocris Biosciences) at a concentration of 2Ki (2nM) prior to infection.

### 2.8. Cramp Elisa

BM PMNs were challenged with *S. pneumoniae* TIGR4 pre-opsonized with 3% naïve or heat-inactivated immune sera at a MOI of 2. Control PMNs were mock challenged with only sera. The reactions were incubated for 40 min at 37 °C. Centrifugation was used to separate the supernatants and cell pellets. RIPA lysis buffer with 0.1% Tx-100 was used to lyse the cell pellets. Cathelicidin antimicrobial peptide levels were determined by ELISA (CRAMP ELISA kit, mybiosource, San Diego, CA, USA, MBS705604), as per manufacturer’s instructions.

### 2.9. Mitochondrial ROS

PMNs isolated from the BM were infected with *S. pneumoniae* TIGR4 pre-opsonized with 3% heat-inactivated immune sera at a MOI of 10. Control wells were mock infected with PMNs and sera alone. MitoSOX^TM^ Red mitochondrial superoxide indicator for live cell imaging (Invitrogen, Waltham, MA, USA, M36008) was then added and the 96-well plate was immediately placed in a plate reader to detect mitochondrial ROS production. Readings were taken every minute over a 1-h time period. Conditions used were an excitation of 485 nm and emission of 585 nm.

### 2.10. Statistics

Statistics were analyzed using Prism 9 (GraphPad). Graphs are shown as mean +/− standard deviation with each data point representing a different mouse. Normality of data was tested via Shapiro–Wilk test before statistical analysis was performed. Significant differences between groups were determined by One Sample *t*- and Wilcoxon tests, and Mann–Whitney test as indicated. *p < 0.05* was considered significantly different and is indicated in figures with *.

## 3. Results

### 3.1. A1 Receptor Signaling Activation Does Not Enhance Intracellular Killing by PMNs from Aged Mice Following Antibody-Mediated Uptake

Previously published work from our lab showed that signaling through the A1 adenosine receptor was necessary to achieve efficient intracellular bacterial killing by the PMNs from young and old naïve mice [22]. To test the role of this receptor in intracellular killing following antibody-mediated bacterial uptake, we performed a gentamicin protection assay with bone marrow PMNs that were isolated from aged mice. In this assay, the PMNs were treated with a specific A1 receptor agonist (2-Chloro-N6-cyclopentyladenosine) or vehicle control (VC). Following the treatment, the PMNs were infected with *S. pneumoniae Δply* TIGR4 opsonized with heat-inactivated (HI) immune sera or naïve sera as a control. To test the intrinsic PMN function independent of the confounding effects of the age-driven decline in antibody responses, sera were collected from young controls. Consistent with prior reports [22], we found that, when *S. pneumoniae* was opsonized with sera from naïve mice, treatment of the PMNs with A1 agonist slightly increased their ability to kill intracellular pneumococcus when compared to vehicle control, and this was statistically significant (Figure 1). However, when *S. pneumoniae* was opsonized with HI immune sera, A1 agonism had no effect on the ability of the PMNs from old mice to kill these bacteria intracellularly (Figure 1). These data show that, unlike in naïve old hosts, the A1 adenosine receptor does not regulate intracellular killing by PMNs following antibody-mediated bacterial uptake in old hosts.

### 3.2. A2B Receptor Signaling Is Detrimental to Intracellular Bacterial Killing by PMNs Following Antibody-Mediated Uptake

To test if other adenosine receptors regulate intracellular bacterial killing following antibody-mediated uptake, we repeated the gentamicin protection assay using PMNs isolated from the bone marrow of young A2AR^−/−^ mice and their wild-type (WT) Balb/c as control. Balb/c and B6 mice have comparable resistance to pneumococcal infection and vaccination against pneumococcal disease is protective in both strains [37,38,39]. As A2AR was previously found to be required for antibody responses to pneumococcal polysaccharides [40], to test intrinsic PMN function, sera were collected from WT controls. We found that there was no difference between the knock-out or wild-type PMNs in their intracellular killing of *S. pneumoniae* when these bacteria were opsonized with naïve or heat-inactivated immune sera, indicating that the A2A receptor was not regulating intracellular bacterial killing (Figure 2). Next, we performed this assay using PMNs isolated from the bone marrow of young A2BR^−/−^ mice and their WT controls (Figure 3). To test the intrinsic function of PMNs, sera were again collected from WT controls. We found that, in the PMNs isolated from young mice, when *S. pneumoniae* is opsonized with HI immune sera, the intracellular killing of these bacteria significantly increased in the absence of A2B adenosine receptor (Figure 3), indicating that signaling through this receptor is detrimental to bacterial killing. A2B receptor signaling did not have an effect on the intracellular killing when the *S. pneumoniae* were opsonized with sera from naïve mice (Figure 3). These data indicate a specific role for A2B receptor signaling in the intracellular killing of pneumococcus following antibody-mediated uptake but not complement-mediated uptake in young hosts.

### 3.3. PMNs from A2BR^−/−^ Mice Upregulate Intracellular Levels of CRAMP Following S. pneumoniae Infection

To begin to understand how PMNs isolated from A2BR^−/−^ mice exhibited improved intracellular killing of pneumococcus after antibody-mediated uptake, we first analyzed the mitochondrial ROS production. Mitochondrial ROS is necessary for pneumococcal killing and it was shown to be elevated in naïve A2BR^−/−^ mice [27]. We performed an assay to quantify the ROS produced by the mitochondria using MitoSOX dye. We found that, following infection with *S. pneumoniae* pre-opsonized with HI immune sera, PMNs from both the WT and A2BR^−/−^ mice upregulated the mitochondrial ROS production to the same extent within the first 20 minutes but that the response was better sustained over time in the WT PMNs (Figure 4A,B). These data suggest that mitochondrial ROS production does not explain the increased intracellular killing we observed in the A2BR^−/−^ PMNs (Figure 3). Next, we analyzed intracellular granule components. PMNs kill intracellular bacteria through several mechanisms, including the delivery of antimicrobial peptides and enzymes to pneumococcus contained in phagosomes [41,42]. Previously, we found that a decline in the intracellular killing of antibody-opsonized *S. pneumoniae* by PMNs from aged mice, when compared to that by young controls, was accompanied by a decline in intracellular cathelicidin-related antimicrobial peptide (CRAMP) [13]. To test if A2B receptor signaling was regulating the intracellular CRAMP levels, we performed CRAMP ELISA on cell pellets collected from WT and A2BR^−/−^ PMNs infected with *S. pneumoniae* opsonized with HI immune sera (Figure 4C,D). We found that, upon infection, there was twice as much intracellular CRAMP in the PMNs from the A2BR^−/−^ mice when compared to WT controls (Figure 4C). When compared to the uninfected baseline, the PMNs from the A2BR^−/−^ mice significantly upregulated the CRAMP levels upon infection (Figure 4D). These data suggest that A2BR signaling inhibits upregulation of the anti-pneumococcal peptide CRAMP in response to infection with antibody-opsonized bacteria.

### 3.4. A2BR Agonism In Vivo Worsens Pneumococcal Disease Outcome in Vaccinated, Young Mice

Since the PMNs from A2BR^−/−^ mice had improved intracellular bacterial killing following antibody-mediated uptake compared to WT controls (Figure 3), we next wanted to test if A2B receptor signaling was important during in vivo infection in a vaccinated host. To test this, young A2BR^−/−^ and WT mice were vaccinated with PCV and, 4 weeks later, infected with *S. pneumoniae* TIGR4 and monitored for bacteremia, their clinical score, and their survival for 7 days post-infection (Appendix A). We found that, when vaccinated, both the WT and A2BR^−/−^ mice were fully protected from pneumococcal disease following infection. We found no incidence of bacteremia in either the WT or A2BR^−/−^ mice (Appendix A), no differences in clinical score (Appendix A), and that 100% of all infected mice survived the infection (Appendix A). As even WT controls were fully protected and therefore improved protection in A2BR^−/−^ mice was not possible to observe, we instead asked if activation of the A2B receptor impairs the protection of vaccinated hosts upon in vivo infection. To test this, young WT mice were vaccinated with PCV and, 4 weeks later, treated with specific A2BR agonist (Bay 60-6583) or VC 18 h before, at the time of, and 18 h post-infection with *S. pneumoniae*. The clinical score and CFU were then analyzed 24 h post-infection. Following infection, the group that received the A2BR agonist had significantly higher clinical scores (Figure 5A) and significantly higher bacterial burdens in the lungs (Figure 5B). These data show that, during pneumococcal infection, the activation of A2BR worsens disease outcome in young, vaccinated hosts.

### 3.5. Absence of A2BR Signaling Does Not Boost Intracellular Killing Following Antibody-Mediated Uptake in Aged Mice

Previously, we found that PMNs from aged mice have a significant defect in their intracellular killing of *S. pneumoniae* following antibody-mediated uptake when compared to PMNs isolated from young mice, which resulted in subpar protection following vaccination [13]. To determine if the A2B receptor can be targeted to improve the bacterial killing by PMNs from aged hosts, A2BR^−/−^ mice were aged for 18 months. PMNs were isolated from the bone marrow and a gentamicin protection assay was used to determine the amount of intracellular killing by WT and A2BR^−/−^ PMNs infected with *S. pneumoniae* opsonized with naïve or HI immune sera. To test the intrinsic PMN function, sera were collected from young WT controls. There was no difference in intracellular killing between WT and A2BR^−/−^ PMNs from aged mice in either of the conditions that were tested (Figure 6). These data suggest that removal of the A2B receptor is not sufficient to boost intracellular bacterial killing following antibody-mediated uptake by PMNs from aged mice.

### 3.6. Inhibition of A2BR Signaling In Vivo Does Not Improve Protection Following Pneumococcal Infection in Aged, Vaccinated Hosts

As activating A2B receptor signaling in young, vaccinated mice worsened the disease outcome, we next asked if A2BR inhibition in vivo in old, vaccinated mice would improve the host resistance to infection. Aged WT mice were vaccinated with PCV and, 4 weeks later, treated with specific A2B receptor inhibitor (MRS 1754) or VC 18 h before, at the time of, and 18 h post-infection with *S. pneumoniae*. The clinical score and CFU were then analyzed 24 h post-infection. When compared to young VC-treated mice, the old VC-treated mice had higher clinical scores and bacterial burdens in the lungs (Figure 5A,B and Figure 7A–C). These data confirmed that aged hosts are not protected from pneumococcal infection despite vaccination, as we have previously reported [13]. When comparing old VC-treated to A2BR inhibitor-treated mice, there was no significant difference in their clinical scores (Figure 7A) or bacterial burden in the lungs (Figure 7B), blood (Figure 7C), or brain (Figure 7D). These data show that the inhibition of A2B receptor signaling in vaccinated, aged hosts does not reverse age-related susceptibility to pneumococcal infection.

As both A2A and A2B receptors are low-affinity adenosine receptors that are G_s_ coupled and have high homology [20,43], we wanted to test if signaling through the A2A receptor would compensate for the A2BR inhibition in vivo in vaccinated aged mice. Aged WT mice were vaccinated and treated with both specific A2B receptor antagonist (MRS 1754) and specific A2A receptor antagonist (3,7-Dimethyl-1-propargylxanthine) 18 h prior to, at the time of, and 18 h post-infection. At 24 h post infection, the clinical score and CFU were analyzed. When both A2AR and A2BR were inhibited, there was still no difference between VC-treated mice and A2BR-inhibition-alone mice in terms of their clinical score (Figure 7A) or bacterial burden in the lungs (Figure 7B), blood (Figure 7C), or brain (Figure 7D). These data suggest that the combined inhibition of A2BR and A2AR in vaccinated hosts does not reverse age-related susceptibility to pneumococcal infection.

## 4. Discussion

Extracellular adenosine signaling is a known regulator of PMN responses to infection and is necessary for host protection against pneumococcal infection. The activation of each of the four adenosine receptors has differing effects on PMN responses. A1 receptor activation has been found to be stimulatory, promoting PMN recruitment to the lungs, adhesion, intracellular killing, and superoxide production [20,21,22,44]. Conversely, A2B receptor activation has been found to be inhibitory, reducing bactericidal activity and inhibiting both PMN recruitment and superoxide production [20,25,27,44]. However, the role of these receptors in immune hosts is not well known. In this study, we found that, unlike in naïve models of infection, A1 receptor signaling does not boost the intracellular killing of *S. pneumoniae* following antibody-mediated uptake by PMNs from aged hosts. In PMNs isolated from young hosts, A2B but not A2A receptor signaling had a role, with A2BR signaling being detrimental to the intracellular killing of pneumococcus following antibody-mediated uptake. However, we found that the removal or inhibition of A2BR was not sufficient to correct the age-driven defects in the antimicrobial activity of PMNs against antibody-bound pneumococci or the age-driven decline in pneumococcal vaccine efficacy. These data indicate that the EAD pathway components controlling PMN responses in naïve and vaccinated hosts are different and distinct and that changes in host aging also effects this pathway.

A1R and A2BR signal differently when activated. A1 is a high-affinity adenosine receptor, meaning lower levels of adenosine are needed to activate it [45,46]. A1R is a G_i/o_-coupled GPCR, meaning that its activation inhibits adenylyl cyclase and inhibits cAMP production [45]. A2BR is a low-affinity adenosine receptor, requiring higher μM levels of adenosine to activate signaling, and is a G_s_-coupled GPCR [43,45,46]. Activation of this receptor stimulates adenylyl cyclase and stimulates cAMP production [45]. Therefore, the activation of A1R and A2BR can have opposing downstream signaling effects, resulting in differences in PMN responses. In prior work from our lab, we found that, when *S. pneumoniae* is opsonized with sera from a naïve, unvaccinated mouse, A1R signaling is required for PMN bacterial killing [22]. Specifically, the intracellular killing following complement-mediated uptake was impaired when A1R was inhibited in PMNs from young mice, and agonism of this receptor in PMNs isolated from aged mice enhanced intracellular killing [22]. Here, we confirmed that agonism of A1R in PMNs from aged mice increased intracellular killing following complement-mediated uptake. However, following antibody-mediated uptake, A1R agonism had no effect on the intracellular killing of pneumococcus in aged hosts. This suggests that A1R signaling is required for the complement-mediated but not the antibody-mediated PMN response. We found, in PMNs isolated from young mice, that, in the absence of A2BR signaling, the intracellular killing following antibody-mediated uptake increased significantly, indicating that A2BR is detrimental to the intracellular killing of antibody-opsonized pneumococcus. Previously, our lab found that A2BR signaling impairs the PMN antimicrobial activity in naïve hosts, where agonism of this receptor impaired the PMN killing of *S. pneumoniae* opsonized with complement, while PMNs from A2BR^−/−^ mice had increased killing [27]. These data indicate that, unlike A1R signaling, A2BR signaling is inhibitory of PMN antimicrobial responses to *S. pneumoniae*, that it is independent of opsonin, and that it impairs both complement-mediated and antibody-mediated PMN responses.

PMNs kill *S. pneumoniae* intracellularly through the fusion of preformed antimicrobial granules with the phagosome and through intracellular ROS production [41,47]. In *S. pneumoniae* infection, ROS produced by the NADPH oxidase complex is not necessary to control this pathogen; however, ROS produced by the mitochondria was required for pneumococcal killing [27]. Previously, it was found that PMNs from A2BR^−/−^ mice had increased mitochondrial ROS production and enhanced the overall killing of *S. pneumoniae* opsonized with sera from naïve hosts [27]. Here, we found that PMNs from young A2BR^−/−^ mice more efficiently kill *S. pneumoniae* intracellularly following antibody-mediated uptake; however, this was not due to higher levels of mitochondrial ROS production. While A2BR is detrimental to bacterial control in both naïve and vaccinated models of infection, these data indicate that the mechanisms of the PMN antimicrobial response between complement and antibody activation differ. This could be attributed to differing signaling pathways that are activated when complement receptors or Fcγ receptors are activated to initiate phagocytosis. PMNs generate ROS through activation of the NADPH oxidase complex by activating GPCRs, complement, and Fc receptor signaling [47]. However, less is known about signaling pathways that generate mitochondrial ROS in PMNs. In macrophages, mitochondrial ROS is generated through the activation of toll-like receptors, proinflammatory cytokines, and calcium signaling [48,49]. In the PMNs, MyD88, a downstream signaling adapter protein of TLR signaling, was shown to be required for mitochondrial ROS production, indicating a role for TLR signaling in PMNs as well [27]. The previously reported increase in mitochondrial ROS by A2BR^−/−^ PMNs infected with pneumococci opsonized with sera from naïve hosts are most likely complement-mediated. While C3b is the component of complement that acts as opsonin, additional complement components have been shown to effect mitochondrial responses in other cell types [28]. Complement receptor C5a was found to be expressed internally on mitochondria in monocytes, and activation of this receptor on mitochondria by C5a induced mitochondrial ROS production in a dose-dependent manner [50]. Complement receptor C3a was also found to localize to the mitochondria and inhibit respiratory function in human retinal pigment epithelial cells [51]. Overall, these findings suggest that additional non-ROS-related mechanisms of intracellular pneumococcal killing are regulated by A2B receptor signaling in young mice.

The fusion of primary and secondary neutrophil granules with the phagosome is necessary for efficient intracellular microbial killing following phagocytosis [42,52]. CRAMP and the human homologue LL-37 are important mediators of pneumococcal killing. We found that intracellular levels of secondary granular component CRAMP were elevated in PMNs from A2BR^−/−^ mice, and infection of these PMNs with *S. pneumoniae* opsonized with HI Immune sera significantly increased the intracellular CRAMP compared to uninfected controls. CRAMP-deficient mice were found to have a higher bacterial burden and mortality during pneumococcal meningitis [53]. Aged mice were found to have prolonged nasal colonization of *S. pneumoniae*, which was associated with diminished CRAMP expression [54]. Additionally, a decreased intracellular killing of pneumococcus by PMNs from aged mice was associated with decreased levels of intracellular CRAMP [13]. In humans, LL-37 was found to be effective at killing different strains of *S. pneumoniae* [55]. Therefore, increased intracellular CRAMP levels could account for increased intracellular killing following antibody-mediated uptake in the absence of A2BR signaling. It is possible that additional antimicrobial granular components play a role in these changes in intracellular killing. Future work will determine if other granular components are involved in these changes, particularly primary granule components which are known to be important for pneumococcal killing such as serine proteases neutrophil elastase and cathepsin G [56,57].

Neutrophils are necessary for protection against *S. pneumoniae* in a vaccinated host, as the depletion of PMNs in vaccinated young mice impaired their resistance to infection [13]. This study identified EAD signaling through the A2B receptor as a pathway that impairs PMN function in young, vaccinated mice. Extracellular adenosine is a damage-associated molecular pattern (DAMP), as it is released upon cellular damage, which can be due to infection and inflammation. Cellular damage significantly raises the adenosine levels from baseline in the extracellular environment and, as a result, signaling through EAD receptors occurs [20]. A2BR is a low-affinity receptor and is activated when adenosine levels are high. Since the adenosine levels rise as infection and inflammation increase, it is possible that A2BR signaling acts as a feedback signal to limit PMN activity, as ROS, NETs, and antimicrobial peptides can contribute to host cellular damage [7].

Following pneumococcal infection, the extracellular adenosine levels in the circulation are higher in unvaccinated aged mice when compared to young controls [21]. Since high levels of EAD activate A2BR, which has been shown to be inhibitory to PMN function, we hypothesized that the inhibition of A2BR in vaccinated aged hosts would rescue the hosts’ susceptibility to disease. However, we found here that the inhibition of A2BR signaling was not sufficient to reverse the age-related decline in PMN function or the age-related decline in vaccine protection against pneumococcal infection. In contrast, our prior studies found that activating A1R signaling was sufficient to improve the responses of PMN to pneumococcal challenge and overall host resistance against pneumococcal infection in aged naïve hosts [21,22]. Our findings here suggest that EAD signaling does not regulate host responses the same way in aged naïve versus vaccinated hosts.

The reasons why targeting A2BR was not sufficient to reverse age-driven defects in PMN responses to antibody-bound bacteria are not clear. Since basal A2BR expression is similar on PMNs isolated from young and aged mice, it is possible that A2BR function is impaired with age to begin with, so further inhibition has no effect [22]. Alternatively, if A2BR activation is similar in young and aged hosts, it is possible that inflammaging (chronic inflammation that occurs with aging) may affect PMN responses despite EAD signaling occurring. Inflammaging results in the increased production of proinflammatory cytokines at baseline, which contributes to immunosenescence and age-related defects in PMN function, as well as the release of immature PMNs from the bone marrow [6,58,59]. While proinflammatory cytokines can stimulate PMN activity, the prolonged stimulation of PMNs can make them less responsive upon interaction with a second stimulus, such as bacteria [60,61]. In support of the role for chronic inflammation in the impairment of PMN responses, PMNs isolated from patients with the chronic inflammatory condition psoriatic arthritis had diminished ROS production, phagocytosis, NETosis, and MPO release when stimulated with TNF compared to healthy controls [62]. Inhibition of A2B receptor signaling may therefore not be enough to overcome the age-related dysregulation of PMN responses induced by inflammaging. Finally, a possible explanation for why the inhibition of A2BR signaling cannot boost the resistance of vaccinated aged hosts to infection is the decline in antibody response that occurs with age. Following both pneumococcal infection and vaccination, antibodies produced by aged hosts have been shown to have reduced opsonophagocytic capacity [11,63]. This change in antibody functionality may result in changes in PMN activation with age, reducing the number of antibodies that interact with phagocytic receptors on the cell surface. Adenosine signaling has been shown to regulate antibody production in response to the pneumococcal polysaccharide vaccine, where CD73^−/−^ and A2AR^−/−^ mice had delayed isotype switching to IgG3 [40,64]. This suggests that impaired adenosine signaling with age may contribute to worsening vaccine responses and that the inhibition of A2BR in vivo may not be able to overcome these age-related defects. In summary, this study has identified a targetable pathway that can boost immune responses in young, PCV-vaccinated mice. This work expands upon the signaling pathways that control vaccine-mediated responses across host age.

## 5. Conclusions

Here, we found that signaling through the A2BR pathway significantly reduced the intracellular killing of *Streptococcus pneumoniae* following antibody-mediated uptake by PMNs from young mice. This was specific to antibody-mediated but not complement- mediated uptake. This change in the intracellular killing by A2BR^−/−^ PMNs was associated with increased levels of CRAMP. Additionally, agonism of this receptor in vivo in young mice resulted in increased susceptibility to pneumococcal infection. However, targeting this receptor in aged hosts had no effect on the intracellular killing or age-related susceptibility to in vivo pneumococcal infection. In conclusion, A2BR signaling at the time of infection is detrimental to the efficacy of the pneumococcal vaccine in young hosts but its inhibition is not sufficient to reverse the decline in vaccine protectiveness in aged hosts.

## Figures and Tables

**Figure 1 vaccines-13-00414-f001:**
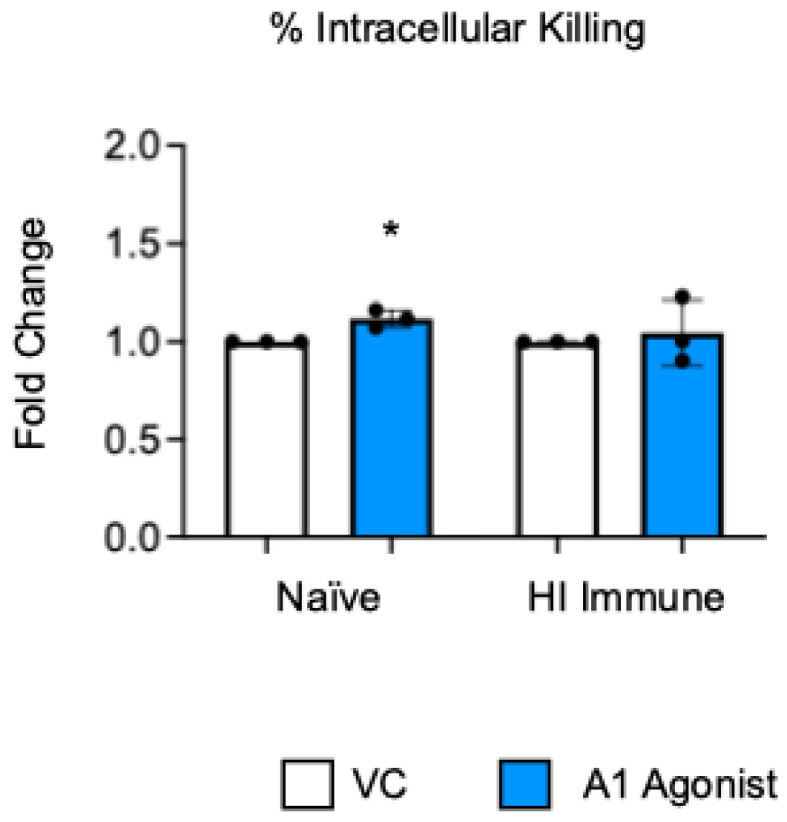
**A1 receptor does not control intracellular bacterial killing following antibody-mediated uptake by PMNs from aged mice.** BM PMNs were isolated from aged C57BL/6JN (20–22 months) mice and pretreated with A1 agonist or vehicle control (VC). PMNs were challenged with *S. pneumoniae Δply* TIGR4 that was pre-opsonized with naïve or HI immune sera. Gentamicin was added to kill extracellular bacteria not taken in. Reactions were plated on blood agar to enumerate CFU. The percentage of engulfed bacteria killed was calculated. Fold changes in bacterial killing of A1 agonist versus VC-treated cells are shown. Data from three individual experiments with n = 3 biological replicates are pooled. * denotes significantly different from one (*p* < 0.05) as determined by One Sample *t*- and Wilcoxon tests.

**Figure 2 vaccines-13-00414-f002:**
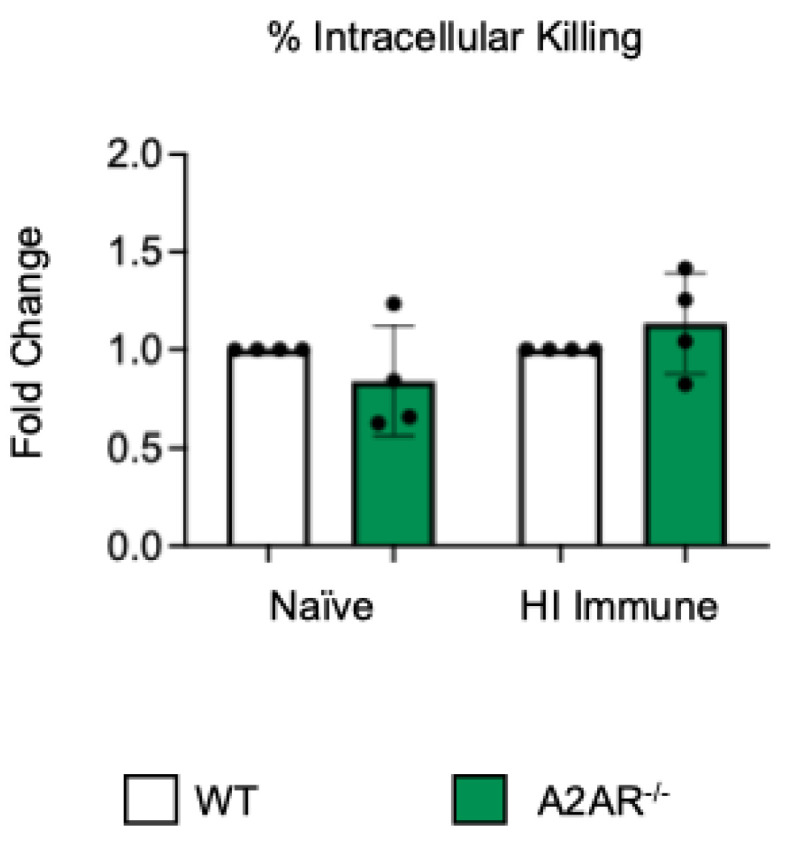
**A2A receptor shows no role in controlling intracellular bacterial killing following antibody-mediated uptake by PMNs**. PMNs were isolated from the bone marrow of young (2–3 months) wild-type (WT) BALB/cJ or A2AR^−/−^ mice. PMNs were infected with *S. pneumoniae Δply* TIGR4 that was pre-opsonized with naïve or HI immune sera. Gentamicin was added to kill extracellular bacteria not taken in. The percentage of engulfed bacteria killed was calculated following plating of the reactions on blood agar to determine CFU. Fold changes in bacterial killing of A2AR^−/−^ versus WT PMNs are shown. Data from four separate experiments with n = 4 biological replicates are pooled.

**Figure 3 vaccines-13-00414-f003:**
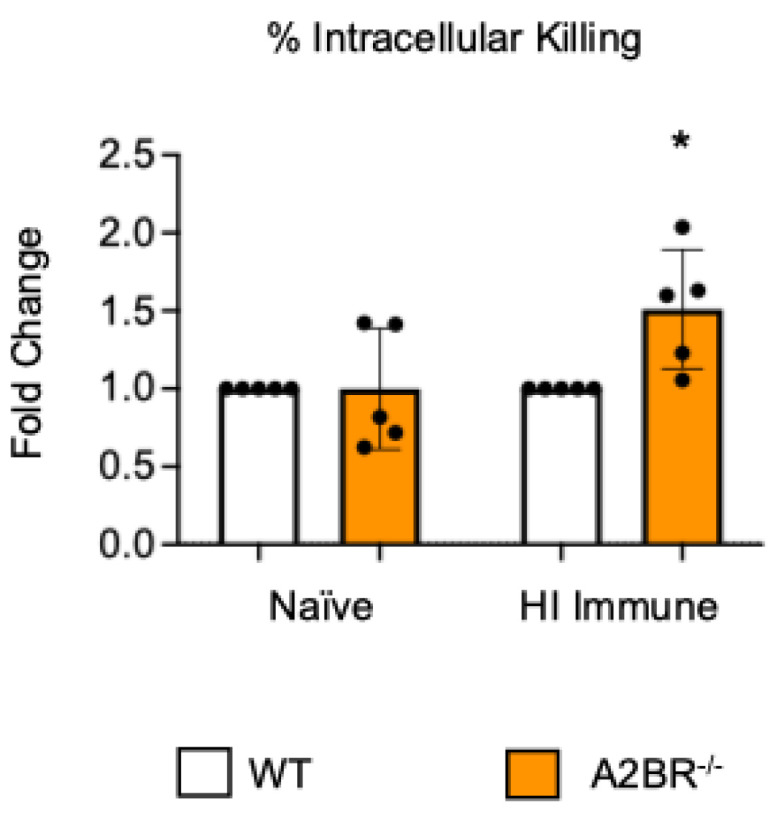
**The absence of A2B receptor increases intracellular killing by PMNs following antibody-mediated uptake.** BM PMNs were isolated from young wild-type (WT) C57BL/6J or A2BR^−/−^ mice. PMNs were infected with *S. pneumoniae Δply* TIGR4 that was pre-opsonized with naïve or HI immune sera. Gentamicin was added to kill extracellular bacteria not taken in. The percentage of intracellular bacteria killed was then calculated following plating of the reactions on blood agar to determine CFU. Fold changes in bacterial killing of A2BR^−/−^ versus WT PMNs are shown. Data from five separate experiments with n = 5 biological replicates are pooled. * denotes significantly different from one (*p* < 0.05) as determined by One Sample *t*- and Wilcoxon tests.

**Figure 4 vaccines-13-00414-f004:**
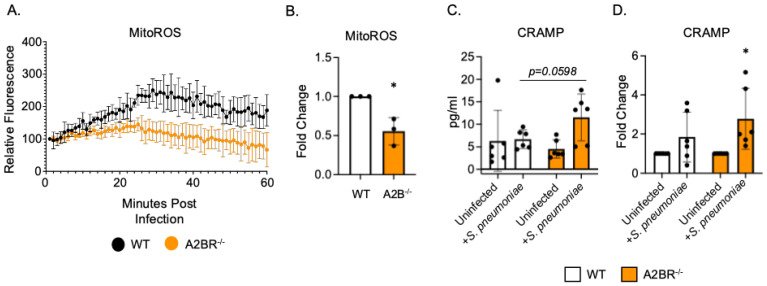
**A2BR signaling blunts intracellular CRAMP levels in PMNs following antibody-mediated bacterial uptake.** PMNs were isolated from the bone marrow of young wild-type C57BL/6J or A2BR^−/−^ mice. PMNs were infected with *S. pneumoniae* opsonized with HI immune sera. (**A**,**B**) Mitochondrial ROS production was determined using MitoSOX. (**A**) Mitochondrial ROS production over the course of 1 hour following infection. Representative data of one of three separate experiments are shown. (**B**) Fold changes in mitochondrial ROS production by A2BR^−/−^ PMNs from wild-type controls were calculated using area under the curve. Data are pooled from separate experiments and * indicates significantly different from one as determined by One Sample *t*- and Wilcoxon tests. (**C**,**D**) Cell pellets were collected and lysed and CRAMP ELISA was performed. Pooled data from separate mice (**C**) and corresponding fold changes from uninfected baselines (**D**) are shown. (**C**) number indicates the *p* value as determined by unpaired Student’s *t*-test and (**D**) * indicates significantly different from one as determined by One sample *t*- and Wilcoxon tests.

**Figure 5 vaccines-13-00414-f005:**
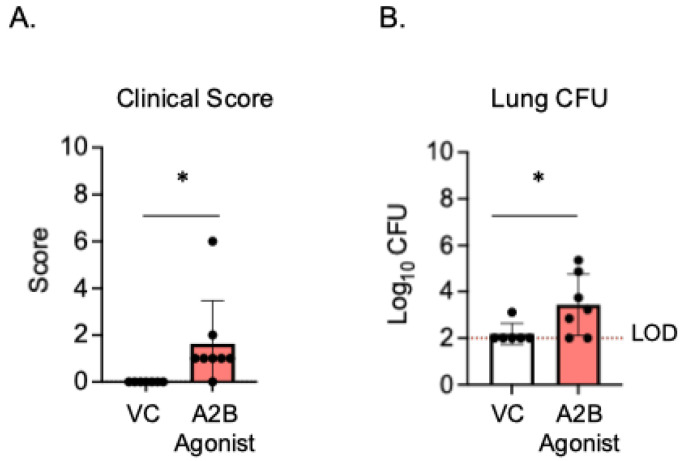
**A2BR activation in vivo worsens disease outcome in young, vaccinated mice.** Young C57BL/6J mice were vaccinated with PCV and, 4 weeks later, infected with 5 × 10^6^ CFU *S. pneumoniae* TIGR4. Eighteen hours prior to, at the time of, and 18 h post-infection, mice were treated i.p with a specific A2BR agonist or VC. twenty-four hours post-infection, mice were assessed for clinical signs of disease (**A**). Organs were harvested 24 hpi and CFU in the lungs (**B**) was determined by plating on blood agar. * denotes significant differences (*p* < 0.05) determined by Mann–Whitney test. The limit of detection (LOD) is denoted by the dashed line. Data are pooled from two separate experiments and each point represents an individual mouse.

**Figure 6 vaccines-13-00414-f006:**
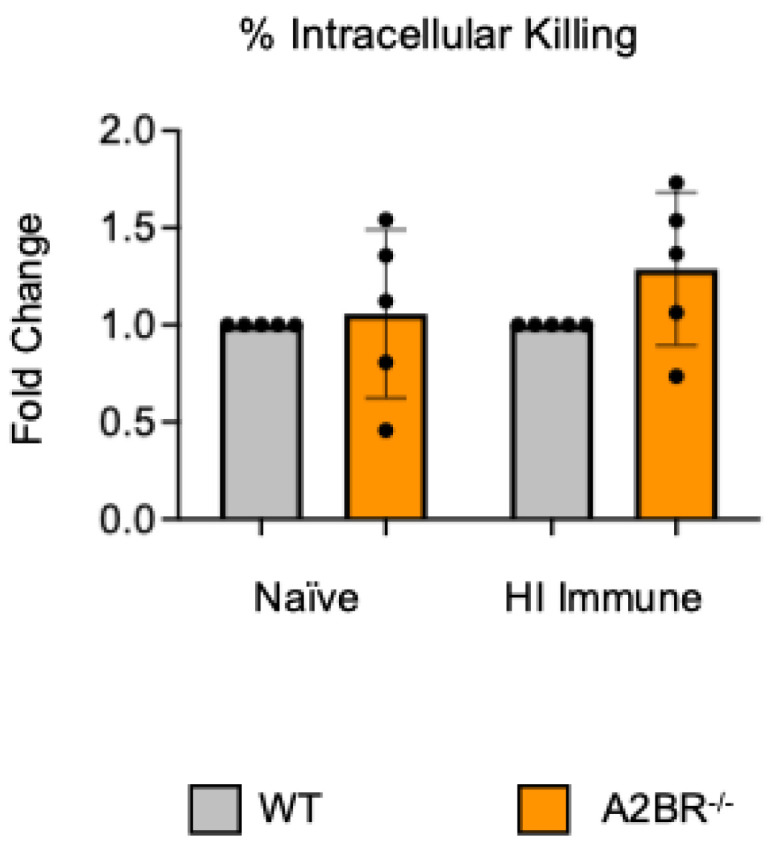
**The absence of A2B receptor signaling does not boost the ability of PMNs from aged mice to kill bacteria intracellularly.** BM PMNs isolated from old (18 months) wild-type (WT) C57BL/6J or A2BR^−/−^ mice were infected with *S. pneumoniae Δply* TIGR4 that was pre-opsonized with naïve or HI immune sera. Gentamicin was added to kill extracellular bacteria not taken in. Reactions were plated on blood agar to enumerate CFU. The % of engulfed bacteria that were killed was then calculated. Fold changes in bacterial killing of A2BR^−/−^ versus WT PMNs are shown. Data from five separate experiments with n = 5 biological replicates are pooled.

**Figure 7 vaccines-13-00414-f007:**
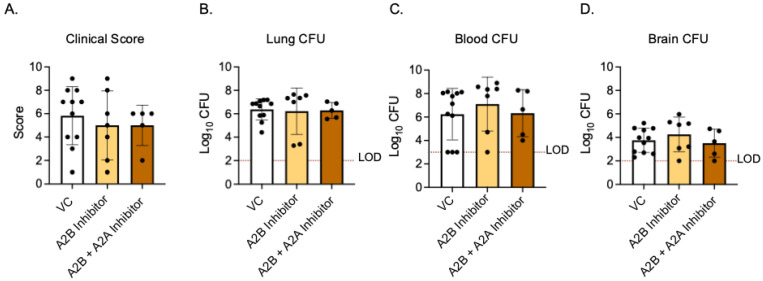
**A2BR inhibition in vivo has no effect on disease outcome in aged, vaccinated mice.** Old (20–22 months) C57BL/6JN mice were vaccinated with PCV and, 4 weeks later, infected with 5 × 10^6^ CFU *S. pnuemoniae* TIGR4. Eighteen hours prior to, at the time of, and 18 h post-infection, mice were treated i.p with either a specific A2BR inhibitor alone, a combination of specific A2BR inhibitor and A2AR inhibitor, or VC. Twenty-four hours post-infection, mice were assessed for clinical signs of disease (**A**). Organs were harvested 24 hpi and CFU in the lungs (**B**), blood (**C**), and brain (**D**) was enumerated by plating on blood agar. The dashed line indicates the limit of detection (LOD). Data are pooled from two separate experiments and each data point represents an individual mouse.

## Data Availability

The raw data supporting the conclusions of this article will be made available by the authors on request.

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
