# Peer review of "Adenosine 2B Receptor Signaling Impairs Vaccine-Mediated Protection Against Pneumococcal Infection in Young Hosts by Blunting Neutrophil Killing of Antibody-Opsonized Bacteria"

_vaccines, 2025, doi:10.3390/vaccines13040414_

Round 1
Reviewer 1 Report
Comments and Suggestions for Authors
In this study, Simmons et al reported that A2B signalling pathway is detrimental to the intracellular bactericidal effect of neutrophils in the presence of specific antibodies opsonising bacteria. While interestingly, targeting A2B pathway shows no effect on the regulation of neutrophil capacity in killing bacterial. In general, this study was well written. My comments are as below:
- the authors described that when bacterial was opsonised by naive sera, treatment of PMN with A1 agonist can significantly increase the killing capacity. But in the figure 1, the data did not look strong enough to support this. Also, in the figure 1 legend which says data are pooled from 4 separate experiments, but only three dots are shown in the bar graph. This should be checked carefully.
- Figure 4, In addition to CRAMP, is it possible that other granule factors within neutrophils also play a role? This should at least be discussed.
Author Response
We would first like to thank the reviewers for their careful consideration and feedback.
Comments and Suggestions for Authors
In this study, Simmons et al reported that A2B signalling pathway is detrimental to the intracellular bactericidal effect of neutrophils in the presence of specific antibodies opsonising bacteria. While interestingly, targeting A2B pathway shows no effect on the regulation of neutrophil capacity in killing bacterial. In general, this study was well written. My comments are as below:
Comments 1: the authors described that when bacterial was opsonised by naive sera, treatment of PMN with A1 agonist can significantly increase the killing capacity. But in the figure 1, the data did not look strong enough to support this.
Response 1: Indeed. we found a 20% increase in intracellular killing that was statistically significant. We adjusted the wording on lines 221-223 to better reflect the data as follows: “treatment of PMNs with A1 agonist slightly increased the ability of these PMNs to kill intracellular pneumococcus when compared to vehicle control and this was statistically significant”.
Comments 2: Also, in the figure 1 legend which says data are pooled from 4 separate experiments, but only three dots are shown in the bar graph. This should be checked carefully.
Response 2: Apologies for the typo. This is now adjusted to 3 separate experiments.
Comments 3: Figure 4, In addition to CRAMP, is it possible that other granule factors within neutrophils also play a role? This should at least be discussed.
Response 3: We added a discussion on other granular components that could be involved on lines 496-500.
Reviewer 2 Report
Comments and Suggestions for Authors
The authors of the manuscript entitled "Adenosine 2B Receptor Signaling Impairs Vaccine-Mediated Protection Against Pneumococcal Infection in Young Hosts by Blunting Neutrophil Killing of Antibody Opsonized Bacteria” presented a well-written study investigating the extracellular adenosine pathway in neutrophil function and protection against infection upon vaccination. This study has a clear research question with some supporting experiments, but there are some points to improve the final quality as follows:
- Introduction: Authors referred to the increased risk of pneumonia in those over the age of 65; however, they didn’t mention the possible reasons for this. Write shortly to explain this point
- The statement “opsonization of pneumococcus” need more explanation
- Materials: Some abbreviations need clarification (full name): RPMI, FBS, EDTA, DMSO, and other abbreviated materials
- Part 3. Unify the font type of the title “Cramp Elisa” like that of the other titles
- Conclusion: There is no conclusion in the manuscript. Add a conclusion part to summarize the most important points and main findings in your study.
- References: The references are written in a wrong format. Return to the author guidelines and correct the format.
- Reference No. 52 is very old (1989). Replace it with an updated one.
Author Response
We would first like to thank the reviewers for their careful consideration and feedback.
Comments and Suggestions for Authors
The authors of the manuscript entitled "Adenosine 2B Receptor Signaling Impairs Vaccine-Mediated Protection Against Pneumococcal Infection in Young Hosts by Blunting Neutrophil Killing of Antibody Opsonized Bacteria” presented a well-written study investigating the extracellular adenosine pathway in neutrophil function and protection against infection upon vaccination. This study has a clear research question with some supporting experiments, but there are some points to improve the final quality as follows:
Comments 1: Introduction: Authors referred to the increased risk of pneumonia in those over the age of 65; however, they didn’t mention the possible reasons for this. Write shortly to explain this point
Response 1: We added a sentence explaining this is due to immunesenescence on lines 38-41.
Comments 2: The statement “opsonization of pneumococcus” need more explanation
Response 2: We now explain what opsonization is in more details on lines 49-51.
Comments 3: Materials: Some abbreviations need clarification (full name): RPMI, FBS, EDTA, DMSO, and other abbreviated materials
Response 3: We added full names for all abbreviations within the manuscript. Apologies for the oversight.
Comments 4: Part 3. Unify the font type of the title “Cramp Elisa” like that of the other titles
Response 4: This was adjusted.
Comments 5: Conclusion: There is no conclusion in the manuscript. Add a conclusion part to summarize the most important points and main findings in your study.
Response 5: A conclusion summarizing major findings is now added.
Comments 6: References: The references are written in a wrong format. Return to the author guidelines and correct the format.
Response 6: We now use the recommended MDPI format.
Comments 7: Reference No. 52 is very old (1989). Replace it with an updated one.
Response 7: We have updated it to a newer one line 511.
Reviewer 3 Report
Comments and Suggestions for Authors
The authors investigated the EAD pathway in neutrophil function and protection against pneumococcal infection upon vaccination across host age. This is an important study to analyze develop a novel strategy to protect elderly people from pneumococcal infection. The investigation is clear and well performed. Specific comments follow.
Major points:
- Line 104, Materials and Methods: Please provide sufficient detail to allow the work to be reproduced, with details of supplier and catalogue number when appropriate.
- Lines 106 & 110: Please make it clear the strain of B6 in the manuscript. Did the authors use both C57BL/6 and 6J strains?
- Line 133: Please indicate the dose and volume of vaccine.
- Lines 150 and 155: Please indicate the volume injected.
- Please describe the sensitivity of B6 and BALB/c mice against pneumococcal infection as well as vaccine efficacy.
Minor points:
- Line 180: “Cramp ELISA” should be “2.8.”.
- Line 187 should be 2.9. and line 194 should be 2.10 then results section should be 3 then discussion should be 4.
- Line 521: Please delete “of role”.
- References should be described as recommended by the ACS style guide. https://www.mdpi.com/authors/references
Author Response
We would first like to thank the reviewers for their careful consideration and feedback.
Comments and Suggestions for Authors
The authors investigated the EAD pathway in neutrophil function and protection against pneumococcal infection upon vaccination across host age. This is an important study to analyze develop a novel strategy to protect elderly people from pneumococcal infection. The investigation is clear and well performed. Specific comments follow.
Major points:
Comments 1: Line 104, Materials and Methods: Please provide sufficient detail to allow the work to be reproduced, with details of supplier and catalogue number when appropriate.
Response 1: We have added sources and catalogue numbers where appropriate.
Comments 2: Lines 106 & 110: Please make it clear the strain of B6 in the manuscript. Did the authors use both C57BL/6 and 6J strains?
Response 2: We receive aged mice for free via our NIA funding. These mice are originally derived from Jackson stock but are housed and bred in the NIA colony at Charles River. The strain is rederived every 6-7 years. Therefore, for those mice we use C57BL/6JN designation. For controls to match the knock-out strains we use C57BL/6J (https://www.nia.nih.gov/research/dab/aged-rodent-colonies). This is now clarified in the materials and methods and is also noted in the legends to differentiate when each is used.
Comments 3: Line 133: Please indicate the dose and volume of vaccine.
Response 3: This is now added to materials and methods.
Comments 4: Lines 150 and 155: Please indicate the volume injected.
Response 4: We describe the volumes on new lines 158 and 162.
Comments 5: Please describe the sensitivity of B6 and BALB/c mice against pneumococcal infection as well as vaccine efficacy.
Response 5: We now describe this on lines 245-246 of the results.
Minor points:
Comments 6: Line 180: “Cramp ELISA” should be “2.8.”.
Response 6: The formatting was corrected.
Comments 7: Line 187 should be 2.9. and line 194 should be 2.10 then results section should be 3 then discussion should be 4.
Response 7: The formatting was corrected.
Comments 8: Line 521: Please delete “of role”.
Response 8: This is deleted from new line 534.
Comments 9: References should be described as recommended by the ACS style guide. https://www.mdpi.com/authors/references
Response 9: We now use the recommended MDPI format.